# MR-Guided Adaptive Radiotherapy for OAR Sparing in Head and Neck Cancers

**DOI:** 10.3390/cancers14081909

**Published:** 2022-04-10

**Authors:** Samuel L. Mulder, Jolien Heukelom, Brigid A. McDonald, Lisanne Van Dijk, Kareem A. Wahid, Keith Sanders, Travis C. Salzillo, Mehdi Hemmati, Andrew Schaefer, Clifton D. Fuller

**Affiliations:** 1Department of Radiation Oncology, The University of Texas at MD Anderson Cancer Center, Houston, TX 77030, USA; bmcdonald@mdanderson.org (B.A.M.); kawahid@mdanderson.org (K.A.W.); klsanders@mdanderson.org (K.S.); tcsalzillo@mdanderson.org (T.C.S.); cdfuller@mdanderson.org (C.D.F.); 2Department of Radiation Oncology, Erasmus Medical Center, 3015 GD Rotterdam, The Netherlands; j.heukelom@erasmusmc.nl (J.H.); dijkvansanne@gmail.com (L.V.D.); 3Department of Computational and Applied Mathematics, Rice University, Houston, TX 77005, USA; soheil.hemmati@rice.edu (M.H.); andrew.schaefer@rice.edu (A.S.)

**Keywords:** MR-guided, adaptive radiotherapy, OAR, normal tissue, head and neck cancer, MRI, quantitative imaging

## Abstract

**Simple Summary:**

Normal tissue toxicities in head and neck cancer persist as a cause of decreased quality of life and are associated with poorer treatment outcomes. The aim of this article is to review organ at risk (OAR) sparing approaches available in MR-guided adaptive radiotherapy and present future developments which hope to improve treatment outcomes. Increasing the spatial conformity of dose distributions in radiotherapy is an important first step in reducing normal tissue toxicities, and MR-guided treatment devices presents a new opportunity to use biological information to drive treatment decisions on a personalized basis.

**Abstract:**

MR-linac devices offer the potential for advancements in radiotherapy (RT) treatment of head and neck cancer (HNC) by using daily MR imaging performed at the time and setup of treatment delivery. This article aims to present a review of current adaptive RT (ART) methods on MR-Linac devices directed towards the sparing of organs at risk (OAR) and a view of future adaptive techniques seeking to improve the therapeutic ratio. This ratio expresses the relationship between the probability of tumor control and the probability of normal tissue damage and is thus an important conceptual metric of success in the sparing of OARs. Increasing spatial conformity of dose distributions to target volume and OARs is an initial step in achieving therapeutic improvements, followed by the use of imaging and clinical biomarkers to inform the clinical decision-making process in an ART paradigm. Pre-clinical and clinical findings support the incorporation of biomarkers into ART protocols and investment into further research to explore imaging biomarkers by taking advantage of the daily MR imaging workflow. A coherent understanding of this road map for RT in HNC is critical for directing future research efforts related to sparing OARs using image-guided radiotherapy (IGRT).

## 1. Introduction

Radiotherapy (RT) treatment of head and neck cancer (HNC) has inspired the development of advanced methods for increased conformity of dose distributions to tumors and organs at risk (OARs) because of the anatomical complexity of the region and its propensity for changes throughout the course of treatment [1]. As the primary target of any treatment planning workflow begins with delivery of the prescribed dose to the tumor, in intensity modulated RT (IMRT) optimizations there are often constraints regarding doses to critical nearby OARs to prevent adverse normal tissue damage as a result of the prescribed treatment [2,3,4]. Limiting dose to OARs is a key concept for optimal treatment outcomes due to inevitable dose deposition in normal tissues surrounding the tumor tissue. The therapeutic ratio is a metric to quantify the optimization problem of balancing tumor treatment while limiting dose to the surrounding tissues. This ratio can be defined as the ratio of the tumor control probability (TCP) to the normal tissue complication probability (NTCP) [5]. Minimizing the NTCP is the focus of many new RT development. Decreased NTCP enable dose escalation studies and limit radiation-induced toxicities that often lead to long term effects such as dysphagia and xerostomia in patients with complete tumor control [6,7]. MR-linac devices present a new opportunity for adaptive RT which can improve NTCP through the use of daily MR imaging. MRI is well established for its superior soft-tissue contrast compared with other imaging modalities that are developing for image-guided RT (IGRT) [8]. Increasing the conformality of dose to the tumor and away from critical soft-tissue structures could lead to long-term reductions in NTCP. Daily MRI also presents an opportunity to improve current NTCP models with the exploration of imaging biomarkers, matching perfectly with adaptive strategies possible on MR-linac devices. This article aims to detail specific methods and developments used in this framework for optimizing the therapeutic ratio in adaptive radiotherapy (ART) by minimizing the NTCP within MR-guided adaptive radiotherapy (MRgART).

## 2. OAR Sparing in Conventional IMRT Planning Process

IMRT was a substantial improvement normal tissue sparing compared to standard 3D conformal therapy, yet further improvement can be achieved as the field aims toward personalized care. The traditional workflow includes a planning computed tomography (CT) that is used for dose optimization. OAR sparing methodologies used in the treatment planning stage depend on the planning strategy used, such as 3D conformal planning, IMRT, and volumetric modulated arc therapy (VMAT) which differ in complexity and time requirements. 3D conformal planning initializes with setting up beam orientations followed by iterative changes to meet required dose constraints. IMRT begins with target and OAR delineations, followed by defining dose constraints and objectives, in order to calculate the dose with a dose optimization algorithm to create the best treatment plan. VMAT adds new degrees of freedom to the IMRT planning approach and allows for beams to be on while the gantry rotates around the patient, creating an ‘arc’ that is highly conformal to the target. IMRT is established to have an increasing conformal dose distribution [9] and aid in OAR sparing [10,11] relative to 3D conformal RT, but some evidence suggests VMAT could further increase the therapeutic ratio [12].

Traditionally, the exact thresholds and constraints used in this inverse planning process are empirically driven values drawing on both the experience of the dosimetrist/oncology team and published reports [4] for general recommendations on dose constraints [13]. These values can vary slightly when optimized on a plan based on patient-specific treatment constraints such as nearness of the tumor to certain structures. 

Following treatment planning, IMRT utilizes a quality assurance protocol (IMRT QA) to verify the complex dose distributions by directly measuring doses with a radiation detector prior to treatment [14]. This step is done to ensure quality treatment and limit inaccurate dose deposition and thus ensures that the delivered dose matches the planned dose. QA also includes the verification and safety steps taken to ensure reliable and repeatable setup of the patient to guarantee optimal placement of the doses given the importance of setup in treatment accuracy [15]. In conventional RT, adaptive workflows require a new CT sim, plan, and QA at each adaptation point and is thus a resource intensive approach.

## 3. MR-Linac Overview

While IGRT using X-ray and CT-based imaging has become routine for clinical use over the last decade, the use of high-field MR-guided adaptive RT (MRgART) has remained unutilized in the US until the FDA approval of the Elekta (Stockholm, Sweden) Unity MR-Linac for treatment in 2018 [16]. Since then, multiple institutions across the world have begun its use for routine clinical practice. Results from its initial implementation in the treatment of oligometastatic, prostate, pelvic, pancreatic, liver, lung, and head and neck cancers are now being published [17,18,19,20,21,22]. The device consists of a closed bore magnet combined with a 6MV Elekta accelerator and allows for simultaneous operation of both applications [23,24]. The improved tumor and OAR delineation allows for adaptive RT during treatment, and the ability to further characterize tissues with quantitative MR imaging biomarkers presents the opportunity for adaptations based on local responses to treatment.

One way the clinical workflow for an MR-linac can differ from conventional therapy is by its use of a combined CT/MR simulation (MR-sim) protocol for delineation of tumors and OARs for treatment planning. Future advances in the generation of Synthetic CTs (sCT) from an anatomical MR image can enable clinics to adopt an MR-only, reducing clinical workflow constraints by eliminating the need to acquire a CT thus simplifying the MR-guided workflow [25,26]. In the on-line treatment workflow, the daily set-up image is registered to the MR sim image to evaluate whether a virtual isocenter shift is needed or the treatment plan needs to be reoptimized. If the plan is reoptimized, then the tumor and OARs are segmented on the daily set-up image, and the adaptive plan dose is calculated. MR-linac devices offer an exceptional opportunity for both increased conformality to treatment targets because of the high contrast daily set up imaging, and the opportunity to explore and integrate quantitative imaging biomarkers into ART workflows which will advance HNC RT to the future of personalized medicine.

## 4. NTCP Modelling

Considering the therapeutic ratio as a primary metric of success in ART, it is crucial to understand NTCP models and how they are used in treatment optimization in ART. NTCP models are mathematical functions that relate the probability of developing a particular side effect to the radiation dose delivered to an OAR. Historically, the Lyman-Kutcher-Burman (LKB) was a common modeling method in NTCP modelling [27]. Nowadays, majority of NTCP models are logistic regression type of models [28,29,30], which can incorporate multiple factors together with dose parameters, such as patient demographics and clinical staging data [31]. NTCP models allow for stratification of patients by estimated toxicity risk, which is currently clinically deployed in the Netherland for the selection of HNC patients for proton therapy [32]. The curve that is produced by an NTCP model represents the estimated risk of a specific toxicity at a specific time point commonly based on a given dose volume histogram parameter for a given OAR and can be used to determine thresholds of risk to inform dose constraints for treatment optimization. Although NTCP models have historically been based on delineations of whole OARs, evidence suggests that sub-regions of the salivary glands seem to have regional differences in dose response [33], which may lead to new treatment planning strategies to reduce the risk of xerostomia in HNC RT. As technological advances in RT treatment planning and delivery allow for more conformal dose distributions, sub-volume NTCP analysis presents an opportunity for more robust characterization of risk to OARs. 

NTCP models could directly be used for dose optimization in adaptive workflows to minimize normal tissue toxicity and improve treatment outcomes. Specifically, NTCP models may be used to determine whether a patient would benefit from mid-treatment adaptive replanning. In a clinical context, this may be done by superimposing the original treatment plan onto a mid-treatment simulation image and calculating dose volume histogram parameters and corresponding NTCP values [34]. If the anatomy has changed sufficiently to increase NTCP above the allowed threshold, then adaptive replanning may be a useful strategy to reduce OAR toxicity for that individual patient.

Dosimetric Impact of Anatomical Changes

One problem with the conventional process for RT is the issue of inter-fractional deformation of the patient’s anatomy [35], which is often attributed to loss of volume in the tumor and changes in weight induced by treatment. In HNC, anatomical deformation can be a considerable problem because of the increased probability of weight loss due to radiation-induced oral sequelae as well as the degree of organ motion caused by weight loss in the head and neck region [36]. With conventional workflows, this problem is not directly addressed, and the patient is treated for the whole RT course with the treatment plan that was based on the pre-treatment anatomy. This may lead a higher delivered dose to the adjacent OARs as the healthy tissue can potentially migrate into area of the original target volume and receive a high dose. Detection of these changes within conventional RT is extremely time intensive as it requires routine imaging that places a heavy burden on clinical schedules. MRgRT provides the opportunity for daily checks of plan quality based on the anatomy at each fraction, which enables on-line treatment plan adaptation to account for anatomical changes and to maintain quality treatment plans that limit unnecessary dose to OARs.

## 5. ART Strategies

ART was originally described by Yan et al. in 1997 [37] as a re-optimize the dose distribution based on the measurements taken as feedback throughout the course of treatment delivery. Since then, there has been broad heterogeneity in the general approach to ART. This heterogeneity produces a broad spectrum of limitations in the possible implementations and decreases overall standardization and inter-site adoption following clinical trials. For example, ART has encompassed a broad range of treatment delivery intents and technique varying from improving patient setup accuracy to re-planning in response to anatomical modifications in the target volumes and OARs [38]. One approach to simplify the adaptive frameworks into an organized nomenclature was proposed by Heukelom and Fuller [39] to classify by the therapeutic intents and implementation strategies for dose adaptation. Figure 1, demonstrates the different techniques produce differing goals for optimization of the therapeutic ratio.

The adaptive strategies aimed at reducing OAR dose include ART_ex_aequo_, ART_OAR_, ART_reduco_, and ART_totale_. ART_ex_aequo_ describes the strategy to maintain planned dose to the target volume and OAR through serial plan verification. ART_OAR_ describes the strategy of reducing the dose to the OARs while maintaining target dose. ART_reduco_ describes the approach of maintaining the planned dose to the target but updating the CTV contour according to the changes in patient anatomy to reduce OAR dose. Finally, ART_totale_ includes an updated CTV shape to conform to the deformed anatomy in addition to an amplified dose to the target volume and reduced OAR dose. All of these methods include an intentional reduction in dose to the OARs and therefore a reduction in NTCP. Theoretically, ART using MRI could be employed to spare OARs based on functional imaging following one of these strategies, but this approach is relatively still unexplored and thus not clinically applicable yet. Possible future applications for these strategies will be discussed in the ‘direction of the technology: quantitative biomarker’ section to follow.

Even within these varying ART intentions, the adaptation interval may still vary depending on the clinical constraints and devices available, as demonstrated by Heukelom and Fuller in Figure 2. In the fixed interval approach, verification imaging is performed at one or more pre-specified time points during RT, and the plan is adapted if anatomical changes are large enough that dose constraints are violated. In triggered adaptation, the plan is adapted when some quantitative (e.g., dose constraint violation, weight loss above a threshold) or qualitative (e.g., poorly fitting immobilization mask) criterion is met. Serial adaptation involves high-frequency (at least weekly) adaptation but does not account for dose delivered during prior fractions. In contrast, cascade ART also involves high-frequency adaptation but updates the accumulated dose after each fraction and uses it in the treatment optimization process. Practically, fixed-interval and triggered adaptation approaches are the easiest to perform on a conventional linac and have been performed in a limited number of prior clinical trials and in silico planning studies [40]. However, without state-of-the-art treatment machines capable of on-line ART such as MR-linac devices, high-frequency adaptation is not clinically feasible due to the time and resource burden on the clinic. The current commercially available MR-linac systems enable serial ART but do not currently have dose accumulation tools for cascade adaptation, although dose accumulation for MRgART is an active area of investigation [41,42].

## 6. Ongoing Phase 2 Studies in HNC

There are several ongoing phase II clinical trials ongoing to explore OAR sparing using MRgART. These include MR-Adaptor, Martha trial and Insight 2.

### 6.1. MR-ADAPTOR—NCT03224000

The Bayesian Phase II Trial of Magnetic Resonance Imaging Guided Radiotherapy Dose Adaptation in Human Papilloma Virus Positive Oropharyngeal Cancer is currently recruiting patients and was initially registered 21 July 2017 [43,44]. The goal of the phase II trial is to investigate dose adaptation impact on locoregional control and normal tissue radiation-induced toxicity by use of MRgRT on a high field MR-linac device for the experimental arm and a standard of care approach for the control arm. Adaptations are according to the ART_reduco_ framework for sparing OARs and reducing the probability of locoregional failure. Symptom questionnaires and video-strobe for vocal cord function are completed each week to monitor radiation-induced toxicities.

### 6.2. MARTHA-Trial—NCT03972072 

The “MRI-Guided Adaptive RadioTHerapy for Reducing XerostomiA” in Head and Neck Cancer (MARTHA) [45] trial is currently recruiting patients and is aimed toward using ARTOAR techniques to reduce xerostomia occurrence in HNSCC patients by use of daily imaging via low-field MRI. The trial contains a single intervention arm for a protocol of daily imaging and once weekly offline plan adaptation and thus follows the fixed-interval ART_ex_aequo_ depicted in Figure 1. Xerostomia evaluation includes objective LENT-SOMA evaluation including flow measurements at baseline, 6 month-, 12 month-, and 24 month-follow up and subjective evaluation using EORTC-QoL questionnaires at the same time intervals. Outcomes of interest for this clinical trial include xerostomia occurrence, locoregional control and overall survival.

### 6.3. INSIGHT-2—NCT04242459

The study entitled, “Optimising Radiation Therapy in Head and Neck Cancers Using Functional Image-Guided Radiotherapy and Novel Biomarkers” is currently recruiting patients and was first posted to clincialtrails.gov on 27 January 2020 [46]. The study includes two parts, one feasibility planning study to consist of 13 patients, and the second part is a single-center, non-randomized, prospective interventional phase I/II study with three independent arms split by disease site or HPV status. The interventional ART_totale_ strategy will be utilized to include a new re-plan at weeks two and four of treatment to account for anatomical changes in patients. The HPV negative oropharyngeal cancer patients who are non-responders will be evaluated for increasing prescribed RT dose which will split from responding patients after ten fractions based on apparent diffusion coefficient (ADC) measurements of the tumor. Part one will produce preliminary feasibility outcomes for the overall study, and part two outcomes include a comparison of overall dose and parotid gland dose between adaptive arms and assesses safety of the dose escalation protocols used.

## 7. Direction of the Technology

### 7.1. Decision-Making Models

As more clinical trials for ART in HNC advance and adaptive strategies becomes more common-place with technological advances to ease the clinical burden, further optimization schemes will become necessary given the array of options clinicians will have in adapting treatment plans. Decision-making processes to incorporate new information gathered with on-line adaptive workflows will be more complex and require investigation. Regarding the initial treatment plan, decisions are made about the segmentation of the OAR and the dose constraints added to the IMRT plan optimization algorithms. Both processes are common areas for implementation of improved decision-making processes, especially with the advent of artificial intelligence in RT [46,47,48] An additional step in the RT workflow that presents opportunities for optimization of decision-making process is the implementation of ART throughout the course of treatment. Decisions of when adaptation will occur and how much to adapt need to be addressed in the ART workflow. Kim, et al. [49] formulated these questions as an optimization problem to be solved through Markov decision processes based on the relative benefit gained for the patient’s cumulative response. Policies within this mathematical framework can then be designed around the adaptive strategy to optimize patient benefit in OAR sparing, through NTCP modeling, and relative cost of re-planning, which cannot be done each day due to clinical workflow constraints. Moving ART towards a variable schedule based on expected benefit to the patient derived from personalized signals from quantitative imaging biomarkers for normal tissue injury characterization represents a step forward for radiation oncology in HNC to limit unwanted toxicities.

### 7.2. Quantitative MRI Biomarkers

Most of the methods for increasing the therapeutic ratio that have been discussed are directed towards improving spatial conformality of the dose distribution around the tumor and minimizing the dose to nearby OARs. Evidence shows that further improvements in outcomes can be achieved using biomarkers to inform on the treatment strategy for both target and OAR dose adaptations [50]. On MR-linac devices, predictive and response monitoring biomarkers present the greatest opportunity for advancement over conventional RT (with CBCT based IGRT) because of the frequency of imaging, high soft-tissue contrast, and functional information. Quantitative MRI (qMRI) is a complex topic due to the multiplicity of possible signal measurement contexts/meanings. In contrast, signal generated in CT imaging is measured in Hounsfield units which correspond to the intensity of X-rays attenuated for a given voxel. Alternatively, MRI signal contrast can depict a multitude of physical properties within quantitative mapping techniques. These can include perfusion and permeability, cellularity, pH, and metabolism, among others. Additional challenges for widespread adoption of qMRI include a dearth of precision and validation studies stemming from a lack standardization [51]. The use of novel pulse sequences such as MR-Fingerprinting [52] and development of improved quantitative phantoms [53] present viable pathways to improve the precision of quantitative techniques and lead to more standardized and validated measurements. The Quantitative Imaging Biomarkers Alliance (QIBA) was created with the directive of establishing such standards to reduce variance and uncertainty of quantitative measurements using MR devices [54]. Despite the variance of these measurements across vendors and pulse sequence parameters, clinical utility of these biomarkers has been demonstrated on multiple accounts, including for OAR toxicity which is discussed in detail in the following sections.

#### 7.2.1. Diffusion-Weighted Imaging

Diffusion weighted imaging is a MRI method to measure Brownian motion of water molecules within tissue and characterize the diffusion as a singular apparent diffusion coefficient (ADC). The ADC biomarker was shown to be useful for prediction of salivary function in response to radiation and could serve as a decision-making tool within an adaptive framework [55]. ∆ADC represents the change from measurement prior to treatment to the measurement taken as some later point. Increases in ADC as a response to radiation is hypothesized to be attributed to increased mobility of water molecules as cells undergo apoptosis and cell walls begin to break down. The ADC is modeled from repeated DWI images at differing b-values and calculated from following relationship.
(1)$$S_{b=X}=S_{b=0}*e^{\left(-b*ADC\right)},$$
where $S_{B=X}$ is the geometric mean of the signal measured for diffusion in the x, y and z directions for a given b-value. With acute changes in ADC early in the treatment regime, DWI is uniquely applicable within MRgRT for the monitoring capability presented within the daily workflows.

#### 7.2.2. Dynamic Contrast-Enhanced (DCE) MRI

DCE MRI provides quantitative information on the kinetic parameters, such as perfusion and permeability, associated with the transient injection of contrast agent (CA) into the imaging volume [56]. By directly measuring tissue perfusion, these parameters, notably Ktrans, are linked with alterations in tissue vascularity, which is a primary correlate with acute vascular injury [57]. Leaky and irregular vascular structures within damaged tissue and malignant tumors lack the typical structure expected in healthy tissue [50]. Osteoradionecrosis is also detectable within DCE perfusion parameters which presents potentially key imaging biomarker candidate for adaptive radiotherapy [58]. Most contrast-enhanced exams are limited to diagnostic and simulation MR scanners, though they may eventually be acquired on MR-linac systems with recent evidence that the gadolinium-based contrast agents are stable under high-energy radiation [59,60].

#### 7.2.3. MR Relaxometry

MR Relaxometry refers to mapping techniques used to acquire quantitative measurements of tissue parameters such as T1 & T2 relaxation coefficients to characterize tissue volumes. MR Relaxometry measurements of radiation-induced injury have been demonstrated in brain [61,62] and liver [63] with evidence of utility within HNC. Improvements in pulse sequence design and additional repeatability validation studies can help integrate these biomarkers safely into the clinical space for use in adaptive workflows. Dose-dependent intensity changes in T1-weighted and T2-weighted images for pharyngeal constrictor muscles have been shown to predict dysphagia in HNC patients undergoing RT [64,65,66]. While these findings are not performed with a qMR acquisition, the signal will likely persist in T1/T2 relaxometry acquisitions and motivate further investigation. This biomarker is hypothesized to arise from the inflammatory reaction in the muscles that causes edema which is linked to the swallowing dysfunction. T1 and T2 MR parameters of the tissue are able to capture this change due to the effect of the microenvironment of the tissue on the relaxation constants. $T1_{\rho }$ mapping has been shown to demonstrate early dose dependent changes in parotid gland tissue for patients undergoing IMRT treatment [63,67]. Considering reductions in dose to parotid gland dose when applicable have been shown to improve quality of life [68], findings such as this are key areas to investigate for qMRI implementations into ART workflows. This change in $T1_{\rho }{}_{}$ contrast within the parotid gland is hypothesized to arise from the development of fibrosis within the radiosensitive salivary gland. Where T1 represents the relaxation time for spin-lattice environments, $T1_{\rho }$ represents the relaxation time for spin-lattice environments in rotational frame of reference [69,70]. This slight difference in pulse sequence acquisition allows for investigation of slower moving biological molecules and their interactions which could explain the measured signal differences throughout treatment in normal tissue due to the fibrosis development in tissue damaged by radiation [71,72].

#### 7.2.4. Radiomics

The general study of high-dimensional quantitative values from varying imaging modalities is commonly referred to as radiomics. Pre-defined features are extracted from the image intensity of a segmented region of interest (ROI) to generate many candidate biomarkers applicable across a broad range of applications. These features include basic descriptions of intensity within the ROI, geometric features related to the shape of the ROI, and texture features composed of higher-order metrics for intensity heterogeneity [73]. Features can be further investigated through image transformations (filters) [74], such as wavelet transforms, which often cause exponential growth of the investigated feature space. Exploration of predictive performance of radiomic features is a current focus in the RT research field, but certain advancements have been made to establish to make radiomic features robust across multiple treatment sites [75]. The vast majority of radiomics studies in HNC are performed in CT, PET, and MRI, as these are the core modalities for diagnosis and RT treatment planning, but additional modalities such as ultrasound [76,77] have also been studied. Ostensibly, any imaging representation that allows for the segmentation of a region of interest can enable a radiomics analysis, highlighting its broad applicability within the RT workflow. Most radiomics studies opt to use the tumor as an ROI, so model outputs are typically related to the tumor in some capacity (i.e., treatment response or prognosis). However, in the context of HNC, treatment outcomes relevant to OARs, i.e., toxicity, have also been explored. Specifically, several studies have used radiomics-based biomarkers to precise acute and chronic xerostomia [78,79,80,81]. In the context of image-guided radiotherapy, some studies have used radiomics to predict adaptive radiation therapy eligibility [82], which could lead to cost and time savings. For comprehensive literature analysis of radiomics in HNC, we refer the reader to the excellent review articles by Wong et al. [83] and Haider et al. [84]. Like previously mentioned quantitative techniques, radiomics is plagued by similar hurdles in standardization and reproducibility [85]. However, a recent push by the imaging biomarker standardization initiative [86] seeks to standardize radiomics definitions to increase reproducibility, thereby facilitating the more seamless eventual transition of these technologies to the clinic. While there is a large degree of optimism for these relatively low-cost methods to mine existing patient images for personalized medicine applications, clinical trials will first need to be run and evaluated to determine the ultimate clinical utility [87] of radiomics. Large phase III clinical trials investigating the applications of radiomics are currently non-existent, but we predict these will increasingly emerge in the future to help more definitely answer the utility of these techniques for patient care. Finally, growing interest in deep learning has started to shift the paradigm of radiomics away [88] from pre-defined ROI-based features to a more end-to-end workflow. This could be particularly attractive for high-volume MR-guided radiotherapy applications, where segmentation of all images may not be feasible or necessary. Importantly, in situations where ROIs are previously segmented, these ROIs may act as additional streams of information in addition to the deep learning defined features which could have an additive effect [89] in predictive model performance.

## 8. Conclusions

HNC present unique challenges for the effective delivery of RT due to the anatomical complexity and propensity for inter-fractional changes in anatomy [35]. The last two decades of research have reiterated the importance of increasing spatial conformality to tumors to limit unnecessary dose to nearby OARs and sparing OARs through an empirical iterative process to optimize dose constraints [4]. This model of improvement may reach a point of diminishing returns as dose distributions become increasingly accurate and online ART is perfected in clinical workflows to adjust for intra-fractional changes. Beyond this course of development is the promise of perfecting the concept of precision medicine by the use of biomarkers, both clinical and imaging based [90]. The exploration of biomarkers is expanding as standardization methods in quantitative MRI develop and noise due to variability in acquisition protocols is reduced, which will lead to future implementation in clinical trials for dose (de-) escalation. MRgRT stands at the focal point of advancements in medical physics technologies and has led to a unique observational capacity of new clinical findings, which provides opportunities for intervention. Further ART strategies will be evaluated as additional quantitative MR biomarkers are explored and understood to optimize RT in HNC. This roadmap for MRgART of HNC is critical for identifying key opportunities to improve survivorship and quality of life.

## Figures and Tables

**Figure 1 cancers-14-01909-f001:**
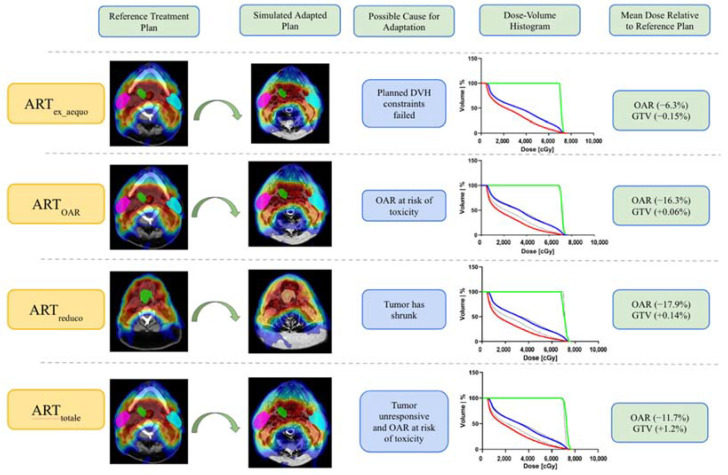
Demonstration of ART intents and relative dosimetric on OARs and tumor. This example shows the initial treatment plan on CT and simulated adapted plans on an MR-linac image for a patient with primary stage T3N2 human papilloma virus positive squamous cell carcinoma of the base of tongue prescribed 70 Gy in 33 fractions. In silico simulated adapted plans were generated with the various ART intents on the MR-linac image from fraction 22. Graphs show dose volume histograms for the gross target volume (GTV) (green), ipsilateral parotid gland (blue), and contralateral parotid gland (red) with solid lines for the adaptive plan and dotted lines for the reference plan. In the column showing the DVH parameters relative to the reference plan, mean dose was used for the parotid glands and D_95%_ for the PTV. For the ART_Reduco_ plan, reduced GTV and PTVs were artificially created by applying a uniform reduction of 1cm in all directions for each structure.

**Figure 2 cancers-14-01909-f002:**
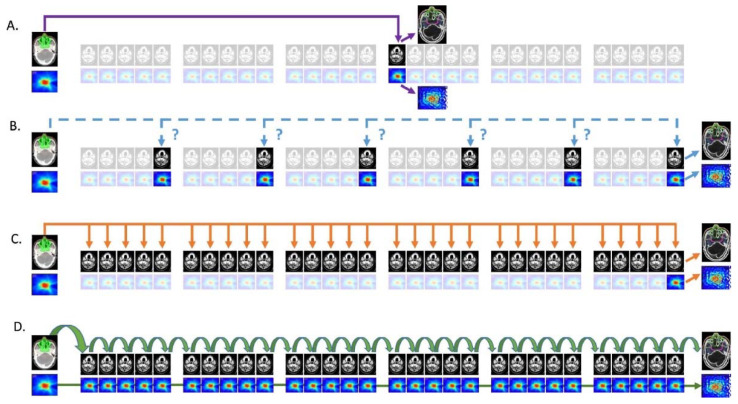
Typologies of ART implementation. (**A**): fixed interval approach; (**B**): ‘triggered’ ART; (**C**): serial ART; (**D**): cascade ART. Figure and copyright permissions obtained from Heukelom and Fuller [37].

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
