# Peer review of "MR-Guided Adaptive Radiotherapy for OAR Sparing in Head and Neck Cancers"

_cancers, 2022, doi:10.3390/cancers14081909_

Round 1

Reviewer 1 Report

The authors provided a comprehensive review on the topic of MR-guided adaptive radiotherapy for OAR sparing in head and neck cancers. The authors are from different institutions with different research and clinical backgrounds. First, the authors compared the traditional IGRT techniques and the MR-guided technique in adaptive planning. MR-guidance has many advantages such as the increased the soft-tissue contrast so as to improve the conformity to treatment targets. The authors also described the NTCP modeling because minimizing the NTCP is the focus of many new RT development. Another issue in H&N cancer treatment is the inter-fractional deformation of patient’s anatomy. Obviously, using the conventional workflow, it will be rather time consuming to detect and monitor the anatomical change during the treatment. But using MR guidance can provide the opportunity to check the anatomical change on the daily base so it will be possible to have the on-line treatment plan adaptation. Later, the authors also discussed the prospect of new technology development. In all, I would like to recommend its acceptance after some minor revisions.

(1). Page 1, line 37 and line 38, please spell out HNC and OAR although you have already done that in abstract.

(2). Line 84. This is the process of “patient specific QA.” “This step is done to ensure quality treatment and limit inac-86 curate dose deposition and thus is an important step in sparing OARs.” This statement is not inaccurate. The patient QA process is not related to sparing OARs. The purpose of QA is just to see if the planned dose distribution from the Treatment planning system agree well with the measured dose distribution in the detector (usually a dose array or multiple dose arrays). I would suggest the author to ask a medical physicists to help review the manuscript’s physics part although this review paper is not physics heavy.

Also, the patient specific QA was not taken by the radiotherapist. At MD Anderson, patient specific QA was done by physics assistants. And in many other hospitals, patient specific QA was done by physics residents too.

“especially when IGRT is not available” I don’t understand why you put it here. The current practice of IMRT all have IGRT. Again, patient specific QA is not related to IGRT. Please ask a medical physicist to proof read this paragraph.

(3). Page 4. Line 174, “is was”, please delete “is”

(4). For the section of MR-LINAC overview, I could not see the introduction of the principle of MR-LINAC. Please ask a physicists to give a brief description of the principle of MR-LINAC.

Reviewer 2 Report

Minor points

#1. Figure 1.

P5L184-185: high-dose planning target volume (PTV) (purple)

>>>I could not find purple lines in Graphs in Figure 1.

#2. Figure 1.

>>>There are several kinds of description in the last columns; OAR and tumor, OAR and GTV, and OAR and Tumor. They should be unified

P6L205. Figure 1.

>>>This should be Figure 2.
